

# Functional network motifs defined through integration of protein-protein and genetic interactions

Amruta Sahoo[1] and Sebastian Pechmann[2]

[1] Département de Biochimie, Université de Montréal, Montréal, QC, Canada
[2] Sebastian Pechmann Research Lab, Saarbrücken, Germany

## ABSTRACT

Cells are enticingly complex systems. The identification of feedback regulation is critically important for understanding this complexity. Network motifs defined as small graphlets that occur more frequently than expected by chance have revolutionized our understanding of feedback circuits in cellular networks. However, with their definition solely based on statistical over-representation, network motifs often lack biological context, which limits their usefulness. Here, we define functional network motifs (FNMs) through the systematic integration of genetic interaction data that directly inform on functional relationships between genes and encoded proteins. Occurring two orders of magnitude less frequently than conventional network motifs, we found FNMs significantly enriched in genes known to be functionally related. Moreover, our comprehensive analyses of FNMs in yeast showed that they are powerful at capturing both known and putative novel regulatory interactions, thus suggesting a promising strategy towards the systematic identification of feedback regulation in biological networks. Many FNMs appeared as excellent candidates for the prioritization of follow-up biochemical characterization, which is a recurring bottleneck in the targeting of complex diseases. More generally, our work highlights a fruitful avenue for integrating and harnessing genomic network data.

## INTRODUCTION

Cells are enticingly complex systems (*Wolf, Katsnelson & Koonin, 2018*). Their phenotypic traits often depend on the concerted action of hundreds or even thousands of individual genes and encoded proteins (*Lehner, 2013*; *Boyle, Li & Pritchard, 2017*). Rationalizing this interdependency is important both for understanding how cellular function is encoded in the genome (*Watanabe et al., 2019*) as well as the origins of complex diseases (*Wray et al., 2018*).

The large-scale mapping of protein-protein interactions (PPI) (*Bork et al., 2004*; *Rual et al., 2005*; *Krogan et al., 2006*; *Tarassov et al., 2008*; *Yu et al., 2008*) and genetic interactions (GI) (*Schuldiner et al., 2005*; *Costanzo et al., 2016*; *Costanzo et al., 2019*) has provided a foundation for understanding the functional organization of cellular networks (*Barabasi, 2004*). Most proteins function through direct contact, *i.e.,* PPIs. Accordingly, clusters of high

Corresponding author
Sebastian Pechmann,
sebastian@pechmannlab.net

interconnectivity in PPI networks have revealed strong hierarchical modularity (*Han et al., 2004*) that ranges from protein complexes (*Gavin et al., 2006*) to sets of functionally related proteins (*Ravasz et al., 2002*; *Huynen et al., 2003*; *von Mering et al., 2003*). Significant efforts have been undertaken to identify such network modules from the network structure (*Park & Bader, 2011*) and link them to observable phenotypes (*Wang et al., 2012*; *Vinayagam et al., 2014*). In contrast, GIs inform on the functional relationship between genes by quantifying epistatic fitness effects of their deletion mutants (*Mani et al., 2008*). Normally, GIs are assessed and quantified through the observed growth phenotype of the deletion of gene pairs compared to the expectation based on single deletions of the corresponding genes (*Beltrao, Cagney & Krogan, 2010*). Positive GIs are observed when the fitness effect of the double-deletion of both genes of interest is less severe than expected based on the phenotypes of the individual deletions. As an example, the first deletion already disrupts a function that requires both genes/proteins, and the second deletion thus has only a comparably small additional effect. As such, positive GIs are often linked to dependencies within the same function, for example between genes of proteins within a protein complex or linear pathway. Conversely, negative GIs are present when the combined effect of deleting two genes is more severe than expected. Negative GIs are frequently observed between redundant and compensatory processes. However, many GIs have complex and non-trivial origins (*Crona et al., 2017*; *Pirkl et al., 2017*; *Kuzmin et al., 2018*) that await to be further characterized. Similar to PPIs, GI networks also display strong hierarchical modularity (*Beltrao, Cagney & Krogan, 2010*; *Cornish & Markowetz, 2014*; *Fang et al., 2019*) that confers robustness and versatility (*Fortuna et al., 2017*).

The smallest yet arguably most important recurring unit of modularity in biological networks are network motifs (*Milo et al., 2004*; *Kashtan & Alon, 2005*). Network motifs were initially defined as small graphlet architectures of size k $\approx$ 3−6 nodes that occur more frequently than expected by chance, thus assuming that their selection indicates importance (*Milo et al., 2004*). Enabled by efficient algorithms for their discovery (*Kashtan et al., 2004*; *Schreiber & Schwöbbermeyer, 2005*; *Wernicke & Rasche, 2006*; *Przulj, 2006*; *Kashani et al., 2009*), network motifs have dramatically improved our understanding of biological networks. Through their detailed characterization it has become clear that motifs often encode important feedback circuits with distinct functionality (*Alon, 2007*) such as feed-forward signaling (*Goentoro et al., 2009*), control of system states (*Elowitz & Leibler, 2000*), and coordination of decision making (*Brandman et al., 2005*). In complement, the large-scale analysis of motifs in networks has helped to connect network topology to function (*Vázquez et al., 2004*), for instance in transcription regulation (*Jothi et al., 2009*; *Wang & Chen, 2010*; *Song et al., 2017*; *Teixeira et al., 2017*), stress responses (*Kim, Kim & Cho, 2012*; *Hahn, Neef & Thiele, 2006*), and development (*Freeman, 2000*).

Importantly, network motifs are primed to encode the evidently pervasive but so far largely elusive feedback and cross-talk pathways within cellular networks required to understand their complexity. However, the usefulness of network motifs to systematically identify regulatory interactions has so far been limited (*Konagurthy & Lesk, 2008*). Specifically, occurrences of motifs are routinely found in very large numbers, which renders the selection of promising candidates for detailed follow-up characterization

challenging. Moreover, despite statistical over-representation of their topology most individual network motifs are not evolutionarily conserved (*Mazurie, Bottani & Vergassola, 2005*), suggesting that only a small fraction of the network motifs may indeed carry function. We therefore hypothesized that an improved identification of network motifs that takes into consideration their individual biological contexts may offer additional insights into the functioning and organization of biological networks.

Here, we defined functional network motifs (FNMs) based on integrating GIs as direct measure of functional relationships between genes into the discovery of motifs in the yeast PPI network. Importantly, only by starting from motifs in the PPI network, FNMs connect to the seminal work on the functioning of network motifs (*Alon, 2007*) and afford to formulate mechanistic hypotheses. This would not be possible to the same extent by starting from GI motifs due to their indirect and complex nature. Remarkably, with occurrences of FNMs about two orders of magnitude less frequent than conventional PPI network motifs, we found FNMs significantly enriched in genes known to be functionally related. Moreover, our comprehensive analyses of FNMs in yeast showed that they are powerful at re-identifying known regulatory interactions as well as discovering interesting new ones. Our work highlights a promising avenue to harness genomic network data towards systematically identifying regulatory interactions and further breaking down biological complexity.

## MATERIALS & METHODS

### Data and code availability

Project data and computer code generated for this study are available at https://www.github.com/pechmannlab/FNM and archived on Zenodo at DOI:10.5281/zenodo.5818964.

### Data sources

The *Saccharomyces cerevisiae* protein interaction network was obtained from BioGRID release 3.5.185 (*Oughtred et al., 2019*) and filtered for physical protein interactions. Yeast genetic interactions were retrieved from *Costanzo et al. (2016)*. The top and bottom 5% of all pairwise interaction scores were considered significant and used to build a genetic interaction network. The set of curated suppressor interactions from *Van Leeuwen et al. (2016)* was used as benchmark example of functionally related genes. We further used the annotations of yeast protein complexes from the MIPS database (*Güldener et al., 2006*), the consensus transcription factors from YEASTRACT (*Teixeira et al., 2017*; *Monteiro et al., 2020*), and the metabolic network model Yeast8 (*Lu et al., 2019*). Co-expression in response to environmental perturbations was assessed with the expression data from (*Gasch et al., 2000*).

### Enumeration of FNMs

Network motifs were generated through exhaustive enumeration of graphlets in the *S. cerevisiae* protein-protein interaction (PPI) network. Our approach followed the algorithm developed by *Kashani et al. (2009)* wherein each motif class of numbers of nodes per layer, defined through the integer compositions of the motif size k, was enumerated by standard

depth-first-search. Herein, starting from a selected source node, all unique patterns of connectivity of a given size were recorded, and all nodes in the yeast PPI network were used as a source node. We considered a graphlet, *i.e.*, a motif in the PPI network, as a 'functional network motif' (FNM) if at least 50% of all possible non-self genetic interaction edges within the graphlet were present, and the source node had direct genetic interactions with all nodes in the most distant layer. The threshold choice of 50% was arbitrary but reflected a best trade-off between sufficient selectivity of a high threshold and sufficient remaining motif occurrences for further analysis. All motifs of sizes $k = 3, 4, 5, 6$ were generated.

Two deliberate constraints served to improve both computational tractability and biological interpretability. Proteins that interact with hundreds or thousands of other proteins play important roles in the organization of cellular networks but rarely exhibit strong specificity and selectivity. To identify functional network motifs that may be involved in specific feedback circuits rather than network hubs, we only considered proteins of degree $d_{max} < 50$, while thresholds of $d_{max} < 25$ and $d_{max} < 100$ were also tested. Importantly, the excluding of the most highly connected network hubs made the exhaustive motif enumeration in the yeast PPI computationally feasible up to a motif size of $k = 6$. Moreover, we only considered interactions between protein complexes but not between subunits of the same protein complexes. The rationale for this was that we wanted to identify putative feedback links that connect biological function rather than understand the architecture of protein complexes that often act as functional unit. Network randomizations were performed with the R package BiRewire (*Iorio et al., 2016*). Motifs were generated for randomized PPI networks, and randomized genetic interaction networks. 30 randomizations were found to yield sufficiently converging reference values (Fig. S1).

## Analysis of FNMs

Motifs were clustered by iteratively identifying and merging maximally overlapping individual motifs. A complex-based motif interaction network was generated by considering all yeast genes/proteins from the FNMs and merging protein complex subunit genes into joint nodes. Genes encoding proteins that are part of more than one protein complex were omitted. Transcriptional co-regulation of genes from the same motif was quantified through their average pairwise cosine similarity between the time-course expression changes in response to environmental perturbations (*Gasch et al., 2000*). All visualization was done in R.

## RESULTS

The identification of regulatory interactions is critically important for understanding cellular complexity. Network motifs have made a foundational impact on our understanding of the regulation of biological networks. Because PPIs and GIs are generally not correlated, the integration of GI data provides a premier opportunity to dramatically expand the functional analysis of network motifs. We here hypothesized that an expanded definition of network motifs based on both PPIs as record of protein activity as well as GIs that quantify functional relationships should improve their usefulness for biological

discovery. To test this idea, we defined functional network motifs (FNMs) as graphlets in the yeast PPI network that were also enriched in GIs. Specifically, for a graphlet to be considered an FNM we required the presence of GIs between the source node and all nodes in the most distant layer, *i.e.,* GIs that span the full graphlet, as well as overall in $\geq 50\%$ of all possible non-self edges (Fig. 1A). The threshold of 50% was arbitrary but reflected our best choice in a trade-off between stringency and sufficient motif occurrences that remained for further analysis (Fig. S1A).

Ultimately, the causality of feedback and regulatory interactions can only be established through experimental perturbation and validation. Our aim here was to provide a computational strategy for the identification of select high-confidence candidates out of the very large number of possible network motifs for future characterization and validation. We here considered as PPI network motifs the collection of all occurrences, *i.e.,* specific sets of genes and encoded proteins, whose wiring architectures were observed at frequencies greater than chance in the yeast PPI. Similarly, FNMs were defined to describe the specific sets of genes and encoded proteins that were found in PPI motifs filtered for GI content.

We implemented an algorithm for the efficient and exhaustive enumeration of all motifs of size $k = 3, 4, 5, 6$ in the yeast PPI network see ('*Methods*'). Two deliberate constraints were imposed to reduce computational cost while increasing biological interpretability. Because the most highly connected nodes in the yeast PPI are usually protein interaction hubs rather than specific regulators, we omitted the highest degree nodes from the network. Moreover, we only considered at most one protein complex subunit per motif as we were interested in links between functional units rather than the architecture of protein complexes. Remarkably, we counted about two orders of magnitude fewer FNMs than PPI motifs independent of motif size (Fig. 1B). While the exact numbers depended on the choice of the threshold, in all cases GIs exerted strong selectivity on the sets of genes and encoded proteins connected within FNMs. For comparison, we generated all PPI motifs in the yeast PPI and randomized PPI networks as well as FNMs upon randomization of the GI network (Fig. S1B, see '*Methods*'). The obtained results followed expected trends. Illustrated for the most populated topologies, motifs occurred more frequently in the PPI network than in randomized PPI networks, thus could also be considered classical network motifs (Fig. 1C). Similarly, FNMs were substantially more frequent than in randomized GI networks, suggesting that the selection of motifs based on high GI density is selective and meaningful (Fig. 1C, Figs. S1C, S1D). Taken together, FNMs collectively identify the same topologies as PPI network motifs but their individual occurrences were substantially more rare.

## FNMs are enriched in functionally related genes

Naturally, a more selective definition of network motifs resulted in lower counts. To test whether our definition of FNMs was indeed meaningful, we evaluated if they preferentially contained functionally related and important genes. One intriguing example of functionally related genes is given by genetic suppressors, pairs of genes where the phenotypic effect of a mutation in one gene is compensated by the other. Importantly, while yeast suppressors

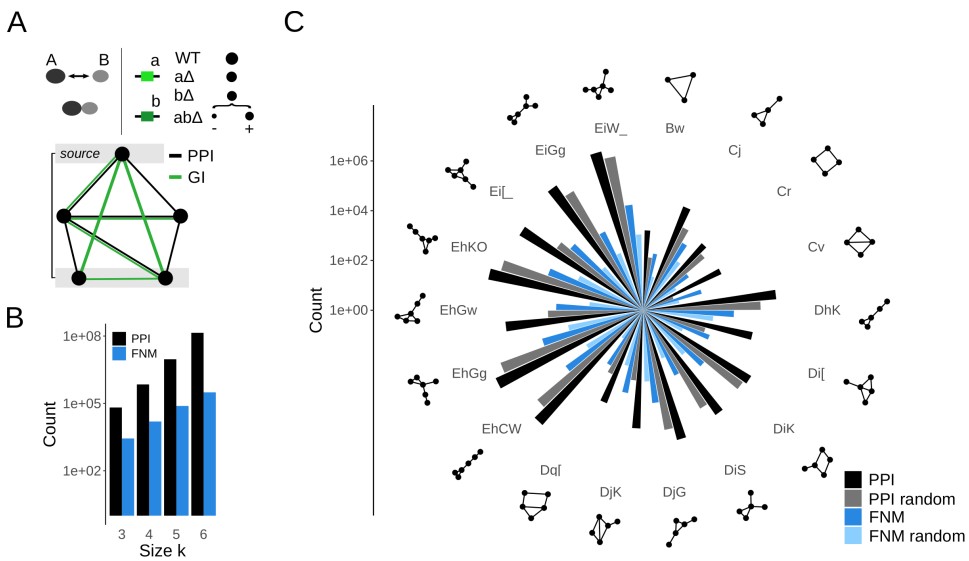

**Figure 1** **Definition of functional network motifs (FNMs).** (A) Exemplary schematic of an FNM. A protein–protein interaction (PPI) between proteins *A* and *B* occurs through direct physical contact of the two proteins and is often functional. A genetic interaction (GI) between the coding genes *a* and *b* is derived from assessing the fitness, normally estimated through a growth phenotype and here schematically illustrated by a circle representing the cell colony size, of *ab*Δ double-deletion strains relative to expectation based on the effect of the single deletions *a*Δ and *b*Δ on cell fitness. Significantly less severe and more severe deviations from expected are recorded as positive and negative GIs. Here, FNMs of pre-determined sizes are generated by integrating the genome-wide PPI and GI networks of *S.cerevisiae*. Black edges represent PPIs and define a graphlet. Green edges represent genetic GIs. The source node as the starting node of the motif search by depth-first-search and most distant layer in the motif are highlighted. A network motif becomes an FNM if GIs are present in at least 50% of all non-self edges, and between the source node and all nodes in the most distant layer. (B) Counts of conventional PPI network motifs and FNMs as function of the motif size k. GIs introduce selectivity and the occurrences of FNMs are about two orders of magnitude lower than that of PPI network motifs. (C) Counts of motifs in the yeast protein and genetic networks for the most represented motif topologies. Shown are the total motif counts based only on the PPI network (PPI), for randomized PPI networks (PPI random), as well as for FNMs and FNMs computed from randomized GI network (FNM random). Motif topologies are indicated both graphically and in compressed graph6 format.

genetically interact (*Van Leeuwen et al., 2016*), a single GI alone would not bias a motif towards an FNM (Fig. 2A). Strikingly, we found a significant enrichment of suppressor interactions in FNMs compared to conventional PPI network motifs ($p < 10^{-16}$, Fisher's Exact Test, Fig. 2A). Thus, FNMs were directly enriched in functional interactions.

Further support for our definition of FNMs was drawn from an analysis of essential genes and genes that code for protein complex subunits. Essential genes are functionally indispensable and their deletion mutants are by definition not viable. However, GIs of essential genes can be measured for instance with DAmP alleles that perturb their expression, and are often pronounced. Similarly, protein complex subunits are important due to the functional importance of the complex as well as the high fitness cost of mutants in individual subunits that can destabilize the whole complex. We found FNMs to contain a higher fraction of both essential genes ($p < 10^{-16}$, Wilcoxon-Mann–Whitney (WMW)

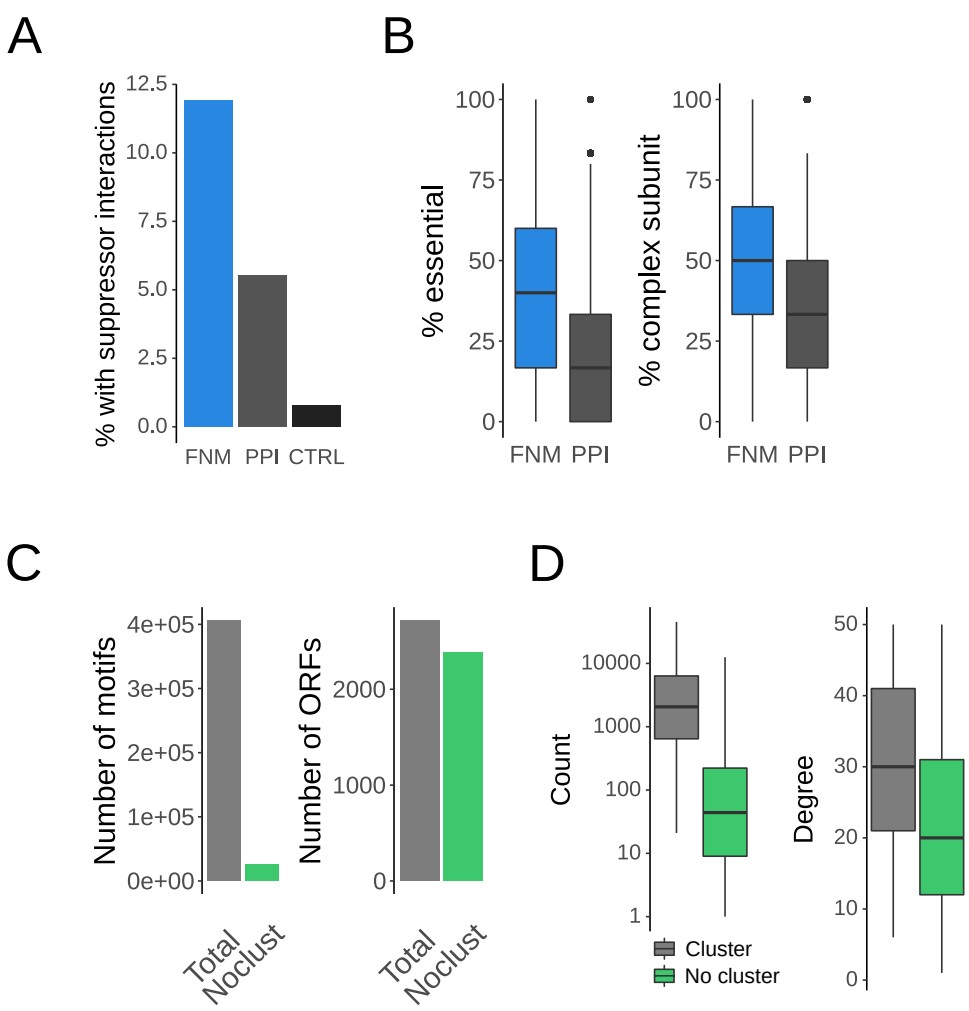

**Figure 2** **FNMs are enriched in functionally important and related genes.** (A) Fractions of FNMs and PPI network motifs (PPI) that contain suppressor interactions as proxy for functionally related genes. As a control (CTRL) are shown the rate of suppresor interactions in randomized motifs with the same content of GIs as FNMs. (B) Fractions of essential genes and genes coding for protein complex subunits in FNMs and in PPI network motifs (PPI). (C) Numbers of FNMs that do not cluster compared to all FNM, together with the number of genes (ORFs) in all FNMs and only in those that do not cluster. (D) Distributions of the number of motifs per gene for the genes in FNMs that cluster *vs.* those that do not cluster together with the network degree for the same groups of genes.

test, Fig. 2B) and genes coding for protein complex subunits ($p < 10^{-16}$, WMW test, Fig. 2B) than PPI network motifs.

To estimate redundancy in the large number of FNMs, we next clustered them based on node overlap see ('*Methods*'). We found that 93.6%, *i.e.,* the vast majority of FNMs concentrated in clusters of sizes $\approx 7 - 70$, while 26078 FNMs did not cluster (Fig. 2C). Of note, very few genes were found responsible for this agglomeration of FNMs. While only 6.4% of FNMs did not cluster, they represented 87.7% of all genes and proteins found in FNMs (Fig. 2C). The genes in clustered FNMs had a significantly higher representation in

different motifs compared to genes only found in non-clustered FNMs ($p = 5.3 \times 10^{-134}$, WMW test, Fig. 2D) because they also had on average a significantly higher degree ($p = 2.3 \times 10^{-30}$, WMW test, Fig. 2D). Importantly, the non-clustered FNMs retained their enrichment in suppressor interactions, essential genes and complex subunits. Thus, FNMs defined through integration of PPIs and GIs are indeed representing collections of important functional interactions. While clusters of FNMs share strong similarity to larger modules in the PPI network, the FNMs that do not cluster should be particularly interesting.

## Genetic interactions in FNMs are orthogonal to protein interactions

Having established that FNMs indeed reflect known functional dependencies between genes, we next sought to better understand the composition of their genetic interactions. Overall, negative GIs were clearly more prevalent in FNMs (Fig. 3A) but we could not identify any trends that link subsets of FNMs to predominantly positive or negative GIs. This was not unexpected as GIs are known to be complex and strongly context dependent (*Markowetz et al., 2007*; *Kuzmin et al., 2018*).

To test any consistency and correlation of GIs and PPIs, we generated a protein complex interaction network by considering all interactions contained in the non-clustering FNMs and mapping them onto joint nodes for the annotated yeast protein complexes. Protein complexes are a good test case because they form important functional units but may interact through multiple subunits. For each pairwise interaction between two complexes, we collected all GIs contained in the corresponding FNMs. While these included both repeated occurrences of the same GI edge from different FNMs and GIs between different complex subunits, almost all pairs of complexes had multiple different GIs. Remarkably, the consensus GIs between protein complexes extracted from the FNMs showed almost perfect consistency towards negative or positive interactions (Fig. 3B, S2).

The full yeast protein complex interaction map highlighted important functional connections as well as the orthogonal nature of PPIs and GIs (Fig. 3C). Some of the most central protein machines required for genome maintenance such as DNA polymerases dominated the observed negative GIs. Complexes with positive GIs included examples of protein targeting and translocation such as the ESCRT complexes involved in endosomal sorting or the Vam3/Vam7 complex that functions in vacuolar trafficking. Only some complexes shared both PPIs and GIs. Notably, only 25.9% of protein complex pairs that interacted physically also interacted genetically (Fig. 3D). In turn, 29.8% of edges between protein complexes were only PPIs, while 44.3% GIs (Fig. 3D). Thus, the GIs captured by the FNMs were both consistent and largely orthogonal to the PPIs.

In the extreme, an FNM of size $k = 6$ can connect six different protein complexes, for instance complexes that function in DNA replication and repair (Fig. 3E). In this exemplary FNM, SLD5 of the GINS complex that is important for DNA replication connected with both PPIs and negative GIs to DPB3 of the DNA polymerase epsilon subunit, DPB11 as scaffolding protein of the Dpb11p/Sld2p complex important for DNA replication initiation, and TOF1 of the replication pausing checkpoint complex. Further connections were observed from DPB11 to MMS4 of the Holiday-junction-resolvase that functions in
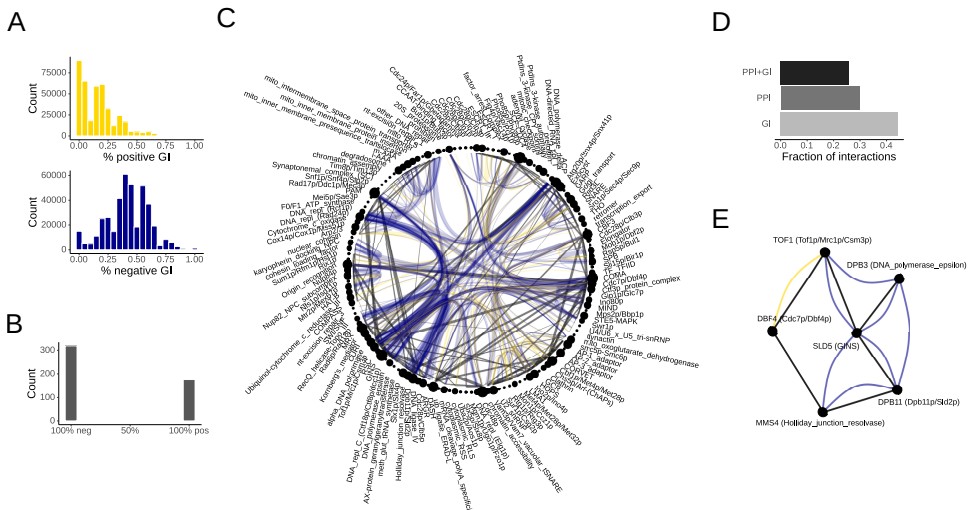

**Figure 3 Genetic interactions in functional network motifs.** (A) Fractions of positive (yellow) and negative (blue) genetic interactions in FNMs. (B) Consensus genetic interactions between protein complexes. (C) Consensus PPI and GI map of yeast protein complexes. PPIs are shown with straight black lines, positive GIs in yellow and negative GIs in blue. The line width reflects the number of interactions observed between the complex pair. (D) Fractions of pairwise interactions between protein complexes that included both PPIs and GIs, only PPIs and only GIs. (E) Exemplary FNM that connects six different protein complexes involved in DNA replication and repair. Both the gene and complex names are indicated.

recombination and DNA repair, as well as a PPI and positive GI between DBF4, which is the regulatory subunit of a kinase complex important for the initiation of DNA replication, and TOF1. The interplay between the regulation of DNA replication and repair is in general well established and this motif was just one example highlighting the potential of FNMs to identify strong candidates for regulatory interactions from genomic network data.

## FNMs identify strong candidates for feedback regulation

Having shown that FNMs capture consistent and meaningful interactions, we next sought to test the potential of FNMs to identify putative regulatory or cross-talk motifs. While some protein complexes come into direct contact, others only communicate through additional proteins with auxiliary roles such as sensing, regulation, or signaling. The betweenness centrality is an established network metric that quantifies the relative fraction of shortest paths that transverse through a node thus indicating its importance for instance for the flow of information across a network. Using the protein complex network generated from merging the FNM interactions of protein complex subunits into joint nodes, we therefore computed the betweenness centrality for all auxiliary nodes, *i.e.,* non-complex nodes, as defined by the shortest paths between all pairs of protein complexes (Fig. 4A).

To test whether the betweenness centrality of the auxiliary nodes identified by the FNMs differed, the corresponding betweenness centralities were also computed for all proteins in the yeast PPI network filtered at maximum degree $d_{max} < 50$, the network used to generate the FNMs. Moreover, we also computed as a control betweenness centrality values for all auxiliary nodes in the full yeast PPI network. Betweenness centrality values
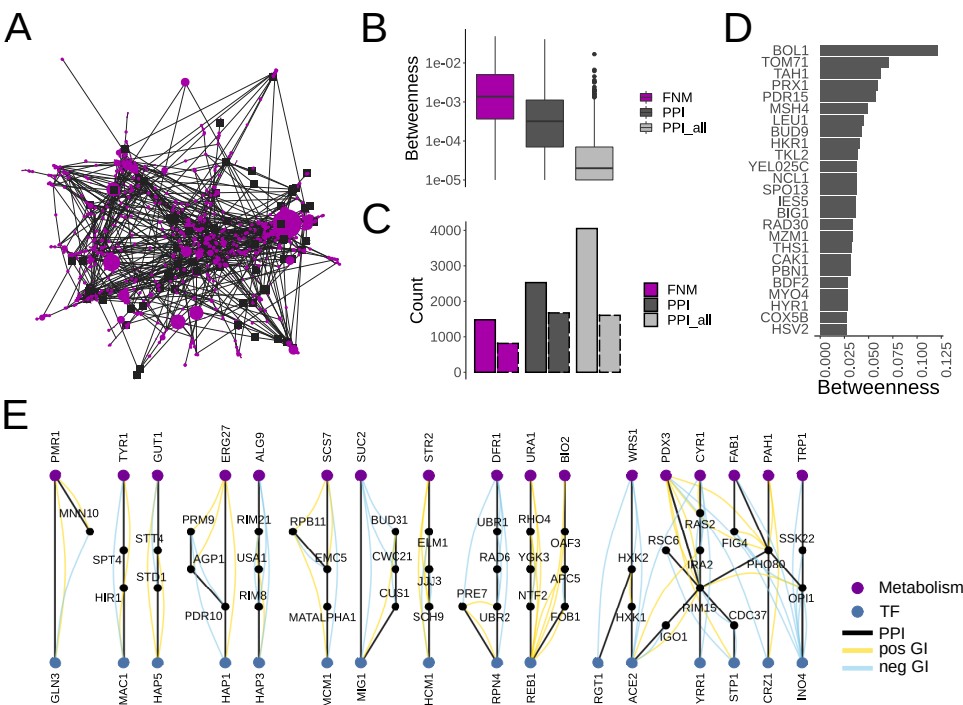

**Figure 4** **FNMs identify known, and novel candidates for feedback and cross-talk interactions.** (A) Yeast protein complex interaction network based on the interactions in the FNMs. Protein complex subunits are merged into joint nodes (black squares). Auxiliary, non-complex, nodes (purple circles) are scaled based on their betweenness centrality. (B) Distributions of the betweenness centrality of the auxiliary nodes connecting protein complexes. Shown are the values for the network derived from the FNMs, from the PPI network with $d_{max}$ <50 as used to derive the FNMs, and from the full PPI network (PPI_all). (C) Network sizes of the FNM, PPI and PPI_all networks. Also shown are the effective network sizes that only include auxiliary nodes with non-zero betweenness centralities (dashed borders). (D) Ranking of the auxiliary nodes with the highest betweenness centrality in the FNM network. (E) All FNMs that connect the yeast consensus transcription factors to the yeast metabolic network.

were significantly higher for FNMs compared to the filtered PPI network ($p = 1.06 \times 10^{-6}$, WMW test, Fig. 4B), and significantly higher for the filtered PPI network than for the full PPI network ($p = 9.24 \times 10^{-174}$, WMW test, Fig. 4B). In part this observation scaled with the corresponding network sizes wherein the shortest paths between protein complexes concentrated to fewer nodes in smaller networks (Fig. 4D). However, considering only contributing nodes with betweenness centrality values >0 suggested that the differences in betweenness centralities for the filtered and full PPI networks was independent of the effective network size and rather a function of the presence of very highly connected network hubs (Fig. 4D). The higher betweenness centrality for the FNM network must be the result of both a smaller network and the lack of hubs.

The investigation of the the auxiliary nodes with the highest betweenness centrality values in the FNM network lead to several proteins known for their regulatory or cross-talk roles (Fig. 4E). BOL1 is a mitochondrial matrix protein and assembly factor involved in the transfer of [4Fe-4S] clusters from the ISA complex to client proteins (*Uzarska et al., 2016*). TOM71 is assumed to aid in the targeting of proteins to mitochondria

(*Schlossmann et al., 1996*). TAH1 is a HSP90 chaperone co-factor required for C/D small nucleolar ribonucleoprotein assembly (*Jiméenez et al., 2012*). The mitochondrial thioredoxin peroxidase PRX1 is important for sensing and countering oxidative stress (*Pedrajas et al., 2000*). And the membrane transporter PDR15 contributes to the cross-talk between stress response and the pleiotropic drug resistance network for cellular detoxification (*Wolfger, Mamnun & Kuchler, 2004*).

To explore potentially novel regulatory interactions in a more controlled test scenario, we next identified all FNMs that connect two of the most important and best-characterized subnetworks in the cell, namely the set of transcription factors (TFs) forming the transcription regulatory network, and the yeast metabolic network. The metabolic network engages in extensive cross-talk to maintain overall energy homeostasis, predominantly through reporter metabolites that bind to regulatory proteins such as transcription factors or metabolic enzymes, but also mediated through PPIs (*Grüning, Lehrach & Ralser, 2010*). Remarkably, we only found 11 FNMs or groups of interlinked FNMs that connect these large and important systems, each with distinct patterns of PPIs and GIs between TFs and metabolic enzymes (Fig. 4E).

The FNMs included the well characterized response to glucose repression where the TF MIG1 directly inhibits the expression of the sucrose hydrolyzing enzyme SUC2. Both were additionally found to genetically interact with the pre-mRNA splicing machinery. The global regulator of respiratory gene expression HAP3 was linked to the enzyme ALG9 involved in N-linked glycosylation in the endoplasmic reticulum *via* two sensors of alkaline pH, RIM8 and RIM21, as well as USA1, the scaffold protein of the ubiquitin ligase HRD1. It is plausible to speculate about a feedback circuit that involves pH sensing and selective degradation. Similarly, the specialized proteasome and stress response TF RPN4 was found to interact not only with the core proteasome subunit PRE7, but *via* the E3 ligases UBR1 and UBR2 as well as the E2 RAD6 also with the metabolic enzyme DFR1 involved in tetrahydrofolate biosynthesis within respiratory metabolism. RAD6 is known to form complexes with UBR1 and UBR2 respectively with different substrate specificities, thus may well contribute to the coordination of RPN4 and DFR1 protein levels. As such, this FNM appeared as an exceptionally strong candidate for contributing to the coordination of transcription regulation and metabolism under stress. In summary, the FNMs have revealed very promising candidates for putative regulatory interactions in low enough numbers that render them immediately amenable for biochemical characterization.

## Transcriptionally co-regulated FNMs are rare but important

Finally, cellular systems are highly dynamic and constantly rewire their networks (*Pe'er et al., 2001*), complexes (*De Lichtenberg et al., 2005*), and also network motifs (*Prill, Iglesias & Levchenko, 2005*; *Doyle & Csete, 2005*) under changing conditions. For instance, dynamically adapting motifs temporally coordinate the global transcription regulation of metabolism (*Chechik et al., 2008*). Without knowledge of the network states under perturbation, the inference of dynamically changing network motifs and their activity under different conditions is challenging and at most approximate. However, the identification

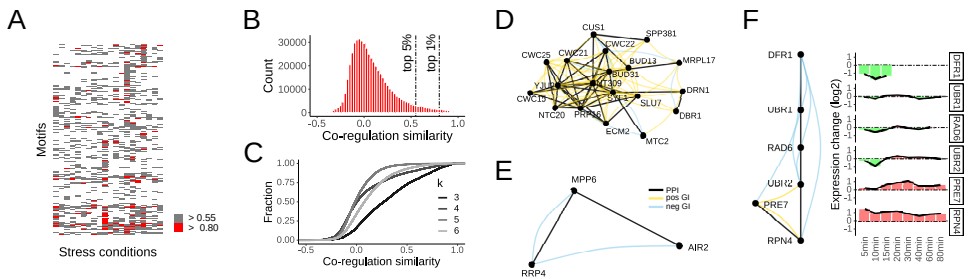

**Figure 5  Transcriptional co-regulation of FNMs in response to environmental perturbations.** (A) Heatmap of co-regulation expressed through average pairwise cosine similarity of the expression profiles of the individual nodes in the motif. The top 5% and 1% are color coded. Rows of motifs with no co-regulation in at least the top 5% are not shown. (B) Distribution of the FNM co-regulation scores. The top 5% and 1% are indicated in the far right tail of the distribution. (C) Cumulative distribution curves of the co-regulation scores as function of motif size k. (D) An exemplary cluster of motifs that is strongly transcriptionally co-regulated during heat stress identifies the core splicing machinery. (E) An exemplary FNM that is strongly co-regulated during heat stress highlights extensive cross-talk between RNA surveillance and quality control. (F) An FNM that connects the transcription regulatory network to the metabolic network may indicate a regulatory function of the E2-E3 complexes formed by RAD6, UBR1 and UBR2 in coupling transcript and protein levels during heat stress.

of strongly co-regulated motifs offered an indication of which motifs retain their activity under changing conditions.

Specifically, by analyzing the transcriptional responses to environmental perturbations we found that co-regulated FNM were rare (Fig. 5A). In fact, less than 1% of FNMs were what can be considered very strongly co-regulated, and about 5% as strongly co-regulated (Fig. 5B). Moreover, while smaller motifs of size $k = 3$ naturally had higher fraction of strongly co-regulated FNMs, co-regulation was else not biased by motif size (Fig. 5C).

Some of the co-regulated FNMs were interlinked and clustered into well-characterized modules, for example the core RNA splicing machinery that was found systematically co-regulated in response to several stress conditions (Fig. 5D). But the small individual FNMs highlighted the more interesting connections. For instance, the nuclear and exosome-associated RNA binding protein MPP6, involved in pre-mRNA surveillance, linked through PPIs to both AIR2, an RNA-binding protein of the TRAMP nuclear RNA surveillance complex, and the non-catalytic exosome core component RRP4 (Fig. 5E). Moreover, RRP4 was interacting through negative GIs with both MPP6 and AIR2, underlining substantial cross-talk between RNA surveillance and quality control (*Aguilar et al., 2020*).

Last, many motifs likely alter their activity under changing conditions. For instance, the previously identified FNM of the stress response TF RPN4 that is linked by an E2/E3 ligase system to the metabolic enzyme DFR1 showed a distinct pattern of expression changes. RPN4 as well as the direct TF target PRE7 were strongly induced under heat stress (Fig. 5F). In turn, the RPN4-facing E3 UBR2 was initially mildly down-regulated, but overall the expression of the E2 and E3 ligases did not change while DFR1 was strongly down-regulated to no longer detected (Fig. 5F). It is plausible to speculate that this FNM contributes to the coupling of transcription and protein levels in adaptation to heat stress by shifting the

protein degradation activity to the RAD6-UBR1 complex for the degradation of DFR1. Of note, while UBR1 is normally nuclear, it readily becomes cytosolic under stress (*Breker et al., 2013*) where it could target DFR1 and other proteins to maintain cellular homeostasis. FNMs are no finitive blueprint of cellular regulation, but offer a most promising glimpse into putative regulatory interactions from already available genomic data that warrant further investigation.

## DISCUSSION

Network motifs have revolutionized our understanding of feedback circuits and the organization of cellular networks. However, solely defined through statistical over-representation they often lack biological context. Here, we have defined functional network motifs, or FNMs, through the integration of genetic interaction data that directly informs on functional relationships between genes and encoded proteins. By capturing known and novel regulatory interactions, FNMs have been found a very promising strategy towards the systematic identification of feedback and cross-talk in cellular regulation. Importantly, the description as 'functional' is intended as categorical rather than definite. Any notion of biological function itself is strongly context-dependent and subject to debate (*Keeling et al., 2019*). More opportunities await for further improving any definition of FNMs.

Ultimately, causation can only be established through targeted perturbations. Large-scale perturbation studies (*McIsaac et al., 2012*; *Hackett et al., 2016*; *Caldera et al., 2019*; *Hackett et al., 2020*) offer a promising outlook. However, even in absence of control perturbations, additional information, especially the directionality of interactions (*Vinayagam et al., 2014*), can be very informative as it allows to explore functional parameter ranges through the modeling of motif dynamics. As more genomic data are generated under non-standard laboratory conditions, it will certainly become easier to dissect the intricacies of cellular regulation.

Already now, FNMs offer many interesting insights and a strategy through the motif-based integration of omics data that is readily extendable to additional data types. Support for this rationale is given by the GIs observed in the FNMs. Similar to other genome-scale measurables, GIs are inherently noisy. Consequently, GIs are routinely analyzed through correlations of their genomic interaction profiles rather than as individual interactions (*VanderSluis et al., 2018*). Through small clusters of GIs between functionally related genes, the FNMs achieve more confidence than would be offered by single GIs but are less generic than genome-wide profiles.

To this end, regulatory events in the cell extend far beyond physical interactions between macromolecules as captured by PPI networks. Signaling can involve feedback through temporal (*Harrigan, Madhani & El-Samad, 2018*) and spatial organization (*Santos et al., 2012*), the dynamic activation and deactivation through modification (*Hirano, Fu & Ptacek, 2016*), and even the folding of proteins and RNA (*Rutherford & Zuker, 1994*). Moreover, some systems are truly determined by collective behavior that can only be understood at a systems level. For instance, the cellular protein folding capacity balances the proteome with available quality control pathways (*Draceni & Pechmann, 2019*) wherein the competition

for the shared folding capacity is governed through differential interaction specificities (*Pechmann, 2000*). Central to many aspects of cell integrity, protein homeostasis is a particularly important example where integrative approaches have been very promising with a lot of open opportunity (*Rizzolo et al., 2017*).

An orthogonal approach to the discovery of critical regulators follows recent advances in control and systems theory to identify the controlling nodes of networks from their topology (*Liu, Slotine & Barabási, 2011*; *Zanudo Tejeda, Yang & Albert, 2017*) and even dynamics (*Baggio, Bassett & Pasqualetti, 2021*). Such avenues currently remain challenging for genome-scale networks, predominantly due to strong assumptions as well as insufficient and incomplete data, but will most certainly continue to gain in importance. Even stronger poised will be the synthesis of these ideas from genomics, systems biology, and control. Due to their simplicity, FNMs can immediately contribute to the systematic identification of cross-talk and regulatory interactions. Iterative cycles of improvement will continue to refine the prioritization of genes for follow-up characterization (*Kuang et al., 2020*), decipher complex regulatory logic (*Buchler, Gerland & Hwa, 2003*), and support the re-engineering of critical protein sense-response systems (*Glasgow et al., 2020*).

### Funding
This research was supported by a Discovery grant from the Natural Sciences and Engineering Research Council of Canada (06504-2016) and the Canada Research Chair in Computational Systems Biology. The funders had no role in study design, data collection and analysis, decision to publish, or preparation of the manuscript.

### Grant Disclosures
The following grant information was disclosed by the authors:
The Natural Sciences and Engineering Research Council of Canada: 06504-2016.
The Canada Research Chair in Computational Systems Biology.

### Competing Interests
The authors declare there are no competing interests.

### Author Contributions

- Amruta Sahoo performed the experiments, analyzed the data, authored or reviewed drafts of the paper, and approved the final draft.
- Sebastian Pechmann conceived and designed the experiments, performed the experiments, analyzed the data, prepared figures and/or tables, authored or reviewed drafts of the paper, and approved the final draft.

### Data Availability
All computer code and project data is available at GitHub: https://www.github.com/pechmannlab/FNM and at Zenodo: 10.5281/zenodo.5818964.

## Supplemental Information

Supplemental information for this article can be found online at http://dx.doi.org/10.7717/peerj.13016#supplemental-information.

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
