# Peer review of "Functional network motifs defined through integration of protein-protein and genetic interactions"

_PeerJ, doi:10.7717/peerj.13016_

## Round 0.1 · original submission · Major Revisions

I would like you to address those concerns from the reviewers.

Reviewer 1 ·

Basic reporting

The authors have neatly described a method to integrate genetic interactions (GI) to define functional network motifs (FNMs) to identify biologically relevant networks. They have critically analyzed the yeast protein-protein interaction network to show the utility of the workflow in identifying known a novel regulatory networks. This workflow provides useful information that needs to be integrated with hub identification from huge and complex networks.

The authors have clearly described the aim and citied multiple relevant articles in the manuscript which needs to be appreciated.

The authors have also provided the codes as GitHub repository, which will be highly useful to other researchers in the field.

Experimental design

The authors have defined FNMs and analyzed the identified FNMs to show their regulatory role. It will be great if the authors can conduct sensitivity analysis, to shown that the defined method identifies biologically relevant networks greater than by chance (eg: AUC).

Validity of the findings

It will be great if the authors can compare their method with other similar methods in the field to show the importance of adding GI.

Additional comments

No comments.

Reviewer 2 ·

Basic reporting

no comment

Experimental design

The research questions are not well-defined. The authors present a new concept that they call Functional Network Motifs or FNMs, and they propose that FNMs reveal new insights into biological networks. However, there are several major shortcomings:

1.) The definition of FNMs appears quite arbitrary, with a threshold of >= 50% frequency of genetic interactions between the genes in a set that is connected by physical interactions. There is no clear justification for this threshold and no analysis of alternate thresholds.

2.) There are confusingly non-parallel definition of FNMs relative to "conventional" network motifs. "Conventional" network motifs are architectures or conceptual graphs that occur more often than expected within a network. In this case, the main significant entity is the motif or architecture itself, not the specific examples. In contrast, FNMs appear to refer to specific sets of genes that are connected by physical and genetic interactions. There is no analysis of whether the observed architectures of FNMs are over-represented other than Figure 1C, but it seems that FNMs are defined as specific example sets of genes, regardless of the prevalence of the architecture. This makes comparisons of conventional network motifs and FNMs not straigtforward.

3) Related to point 2 above, a critical feature of conventional network motifs is the directionality of edges. This is particularly important for interpreting feedback or feedforward wiring. This feature is absent from protein-protein interaction and genetic interaction networks. As such, it is difficult to identify motifs that can be interpreted as relating to feedback or feedforward connections. Identifying such connections appears to be a major goal of the authors, but it is not clear that FNMs are a good way to identify them, and the claims that FNMs are useful for identifying them are not tested appropriately (see below).

4) As a smaller point relating to point 2, the FNMs are defined in terms of a "source node" but the definition of the "source node" in terms of identifying FNMs is not clear (are all nodes in each connected subgraph of the network considered as potential source nodes?). It is also not clear if the term "source node" is necessary or meaningful for FNMs.

The major claim of the manuscript is that FNMs are useful for identifying new functional relationships between genes or protein complexes, especially feedback relationships. These claims are not tested directly, and it is not clear if FNMs actually provide added value over what can be inferred directly from the original PPI and GI data sets.

5) The subsection titled "FNMs identify strong candidates for feedback regulation" and the phrase in the abstract "suggesting a promising strategy towards the systematic identification of feedback regulation in biological networks" suggest that FNMs have identified new feedback relationships that have been validated, but this is not the case. Instead, this subsection describes several FNMs and the relationships between the included genes based on previously published results. Potential regulatory connections are suggested, but the predicted feedback relationships are not concretely defined and they are not tested. Therefore it is totally unclear whether they are indeed feedback relationships. Furthermore, there is no roadmap presented for how feedback relationships might be systematically predicted based on FNM definitions.

6.) In Line 237, the authors comment that genetic interactions between complexes are almost perfectly consistent across the members of the complex. This is an observation that has been made previously by others, but it appears to be presented as a new finding. I believe the earliest description of this feature (relating in this case to pathways rather than complexes) was in Segre et al Nat Genetics 2004. It was also described in Roguev et al Science 2009, and in others as well.

Validity of the findings

1. The authors state that it is remarkable that FNMs are two orders of magnitude less frequent than "conventional" network motifs, but the FNMs are defined with a threshold for frequency of genetic interactions which makes them less frequent than the conventional motifs by definition. The frequency of FNMs is entirely dependent on this threshold, and lower thresholds or higher thresholds would result in FNM definitions with either greater or lower frequency.

2. The statistical test for enrichment of suppressors (line 193) would need to account for the fact that FNMs are enriched in GIs by definition. The authors do not account for this, stating "a single GI alone would not bias a motif towards an FNM". While a single GI will not be sufficient to make a set of genes meet the requirement for being an FNM, the reverse is clearly true. If a set of genes is an FNM, then it is by definition enriched for GIs. It then follows that they should be enriched for suppressor GIs as well. To demonstrate a true enrichment of suppressor GIs beyond what would be expected just from the definition of FNMs, the authors would need to perform a statistical analysis that corrects for the elevated frequency of GIs in FNMs. Is the fraction of GIs that are suppressor GIs higher within FNMs?

Additional comments

In summary, it is not clear whether the definition of FNMs proposed by the authors reveals new biological connections or facilitates the identification of feedback connections.

·

Basic reporting

The authors propose an approach for selecting graphlets from PPI networks that also have genetic interactions. In this case, two genes are defined to have a genetic interaction when the alleles of variations in one of the genes are correlated with the alleles with the variations in the other gene. It is assumed that the correlated variations indicate relationships such as one variation compensating for another to maintain function.

- Language is clear and unambiguous
- Literature references are comprehensive, relevant, and appropriate
- Article structure, figures, and tables are all professional and high quality

On lines 56-57, the authors use the term "deletion mutants." Do they mean "deleterious mutations"?

On line 195, did you mean "FNMs" instead of "FNM"?

Experimental design

- Research question is well defined, relevant, and meaningful. The contribution and its relationship to previous work is clear
- The analyses were rigorous and conducted to a high technical standard
- Methods are described in sufficient detail

Validity of the findings

- The conclusions are well supported.
- The statistical analyses are clear.
- Conclusions are well stated.

Additional comments

Thank you for the opportunity to read about your work. Your approach is interesting and compelling.

I think the clarity of manuscript could be improved for non-experts with a descriptive figure. In particular, I would propose that the authors create an additional figure demonstrating a graphlet identified from the PPI and which edges also count as genetic interactions. The figure could also illustrate why the genes encoding a pair of proteins are classified as having example a genetic interaction (e.g., show multiple sequence alignments for each gene showing the variations across samples). This is completely optional, however -- I would not require this for acceptance.

---

## Round 0.2 · Minor Revisions

Please ensure that all review comments are addressed. As pointed out by reviewer 2, it will be better to spend more time on PPI data interpretation.

Reviewer 1 ·

Basic reporting

The authors have made great effort to address most of the concerning points, therefore would like to appreciate them.

Experimental design

The authors have addressed my comments efficiently.

Validity of the findings

No comments

Additional comments

No comments

·

Basic reporting

- There is a typo in the legend for Figure 1a. ("recoreded")
- I appreciate the addition of figure 1a to clarify some of the terminology and concepts.

- I appreciate the additional explanation on lines 57-62. That said, I think the language used by the authors is still confusing, however.

For example, your paper states: "Positive GIs, wherein the double deletion is less severe than expected, are often linked to dependencies within the same function while negative GIs,
derived from deletions of two genes whose combined effect is more severe than expected, are
frequently observed between redundant and compensatory processes (Beltrao et al., 2010)."

The source you cite explains the same concepts differently: "We have shown that pairs of yeast mutants that display positive genetic interactions often indicate two proteins that act in the same pathway or are physically associated (Figure 1B) (Roguev et al., 2008, Collins et al., 2007). A possible explanation is that if removal of a component of a complex disables that complex, then deleting a second component would have no additional effect, resulting in an epistatic (i.e., positive) interaction (Figure 1A). Alternatively, deletion of one component of a complex could result in partial dysfunction of that complex with a detrimental effect on cell viability. If the removal of an additional component completely disabled this detrimental function, then the result would be a suppressive relationship, another type of positive interaction."

I think the explanation provided by the source (e.g., the impact of deleting the pair is not much greater than deleting a single gene) is much clearer than the language used by authors. Since the paper is focused on the functional nature of the interactions, the authors should expand their explanation. Genetic interactions are very important concept to understanding the overall paper.

- Another reviewer commented on the confusion between the terms "conventional" network motifs and FNMs. I think it would be clearer to label them as PPI network motifs and PPI+FNM network motifs. All of the motifs you find are conventional PPI network motifs -- if I am understanding correctly, you are just selecting the subset for which a minimum fraction of the edges have GIs.

Experimental design

no comment

Validity of the findings

- It seems to me that you could have tried a few different approaches. For example, you could generate a GI network and filter the motifs by a minimum fraction of PPIs. Why is it advantageous to start with the PPI network and filter by GIs rather than the other way around? PPIs capture direct interactions of adjacent components of a pathway, while GIs capture indirect relationships (such as two genes that serve in redundant pathways). Please spend more time clarifying these differences and how the presence of a PPI but not GI, GI but not PPI, or both should be interpreted.

Additional comments

High level comment: I think this paper is well suited to an audience familiar with PPIs or GIs who are looking for new analysis methods. Explanations of the fundamental concepts and how to interpret the results are insufficient for a generalist bioinformatician to decide whether FNMs would be a useful analysis to do for their given problem.

---

## Round 0.3 · accepted · Accept

Please make sure you've checked the instructions for authors about the figure resolution and formatting requirements.